# PERSPECTIVE

## Treated mothers, troubled hearts: cardiovascular trade-offs of metformin use in pregnancy

Clare M. Reynolds

*Discipline of Dietetics, School of Public Health, Physiotherapy and Sports Science/Conway Institute/Institute of Food and Health, University College Dublin, Belfield, Dublin, Ireland*

Email: Clare.reynolds@ucd.ie

Handling Editors: Laura Bennet & Christopher Lear

The peer review history is available in the Supporting Information section of this article (https://doi.org/10.1113/JP289360#support-information-section).

### Cardiovascular outcome in 12-month-old male and female offspring of metformin-treated obese mice

Josca Mariëtte Schoonejans, Phoebe Wilsmore, Lais Vales Mennitti, Kwun Kiu Wong, Thomas J Ashmore, Tessa A.C. Garrud, Heather Louise Blackmore, Olga Patey, Denise S Fernandez-Twinn, Dino A Giussani and Susan E Ozanne

The impact of gestational exposures on the lifelong health trajectory of both the mother and her offspring has captured the attention of the research community. Maternal factors such as diet, stress, environment and maternal health status can predispose the next generation to a myriad of negative health consequences, including cardiometabolic disease. It is therefore not surprising that there is growing interest globally to ensure the health of future generations by optimising maternal care before and during pregnancy.

There is an increasing rate of reproductive-aged women living with overweight or obesity, increasing the likelihood of complications during pregnancy. Gestational diabetes mellitus (GDM), a form of diabetes that develops during pregnancy, is one of the most common complications of pregnancy worldwide, affecting approximately 1 in 7 women. GDM is associated with increased risk for macrosomia and stillbirth and increased incidence of cardiometabolic disease in both mother and offspring in the longer term. The first line of treatment involves lifestyle modifications such as dietary changes and moderate activity (Plows et al., 2019). However this is not always sufficient, and often pharmacological interventions are required to achieve adequate glucose regulation. Insulin remains the standard treatment, but there has been an increase in the use of the oral biguanide, metformin (Guo et al., 2019). Notably several countries, including the UK, have incorporated metformin as a first-line pharmaceutical treatment. In general metformin use is more convenient (oral rather than injection), is associated with less maternal weight gain and often delays or reduces the need for insulin treatment. However concerns remain regarding placental transfer and uncertainty about long-term health outcomes in offspring.

Human studies investigating the postnatal effects of maternal metformin exposure have yielded mixed findings. Some studies report reduced risks of neonatal hypoglycaemia and preterm birth with additional beneficial effects in the offspring of women with polycystic ovary syndrome (PCOS). However children exposed *in utero* to metformin have shown increased adiposity, higher BMI, larger waist circumferences and trends towards elevated glucose and blood pressure by mid-childhood (Hanem et al., 2019). In some cases these effects appear to be sex-specific, with worse effects seen in male children. Given the ethical and logistical limitations of long-term human studies, there is a critical need for animal models to elucidate the effects of prenatal metformin exposure into adulthood.

Schoonejans et al. utilise a well-established mouse model that mimics maternal obesity and gestational glucose intolerance to study the longitudinal effects of metformin exposure during pregnancy on offspring cardiovascular health. This provides valuable insights into potential age- and sex-specific effects equivalent to middle age in humans. Previous work has shown that maternal metformin treatment during obese pregnancy reduced maternal fat mass and body weight in late gestation but did not improve key pregnancy outcomes such as fetal growth restriction or litter size, highlighting a disconnect between maternal metabolic improvement and fetal benefit (Schoonejans et al., 2021). Building on these findings this study shows clear evidence that *in utero* exposure to maternal obesity, with or without metformin treatment, leads to long-term cardiovascular changes in offspring.

Systolic blood pressure (SBP) is a readily measurable and overt indicator of cardiovascular function, often reflecting underlying structural or functional changes within the vascular system. Female, but not male, offspring from the obese group had increased SBP and aortic tunica media thickness throughout the life course. This correlated with other cardiometabolic measures such as adiposity, hyperleptinemia and hyperinsulinaemia.

Further investigation determined that several factors may have influenced the increased SBP and aorta remodelling in these female mice. Examination of the femoral artery revealed heightened vasoconstrictor reactivity and increased vascular resistance, suggesting underlying endothelial dysfunction and altered vascular tone as contributing mechanisms. In addition to these peripheral vascular effects echocardiographic analysis revealed early signs of diastolic dysfunction, including a prolonged isovolumetric relaxation time (IVRT) and increased myocardial performance index (MPI), suggesting impaired cardiac relaxation despite preserved systolic function. Interestingly there was no correction of SBP or femoral artery vasoconstrictor activity with maternal metformin treatment during pregnancy. Although IVRT and MPI changes were not observed in the metformin group, there was evidence of increased diastolic dysfunction with impaired relaxation (E/A ratio), although this normalised by 6 months of age.

Similar to other studies examining the implication of maternal obesogenic diet during pregnancy, sex-specific effects were observed. In male Ob offspring there was no evidence of altered SBP or aorta remodelling. However subclinical effects were observed with increased cardiac output at 3 months and increased E/A ratio (increased blood entering the ventricle under filling) signalling a shift towards diastolic dysfunction. This diastolic dysfunction persisted and was progressive at 12 months of age. Metformin exposure during pregnancy did not prevent these effects and was associated with worse diastolic dysfunction, cardiomegaly and fibrosis. Additionally Ob-Met males showed

vascular hyperreactivity to phenylephrine, which may signal underlying endothelial dysfunction.

In summary this study underscores the importance of considering sex-specific outcomes in developmental programming research, as male and female offspring exhibited distinct cardiovascular phenotypes in response to maternal obesity and metformin exposure. The findings also highlight the subtlety of these programmed effects, which may remain subclinical in early life but could be unmasked or exacerbated by later-life environmental 'second hits' such as a high-fat diet. A notable limitation of the study is the absence of an insulin-treated comparison group, which would be essential for determining whether metformin confers greater or lesser long-term risk than the current standard of care. Crucially this work reinforces the need to study offspring, especially in a sex-stratified manner, when evaluating the safety and long-term consequences of gestational metformin use. It also advances our understanding of how maternal health status and pharmacological intervention interact to shape offspring cardiovascular trajectories, revealing that metformin may exert effects beyond those attributable to maternal obesity alone.

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

## Additional information

### Competing interests

The authors declare no conflict of interest.

## Author contributions

Clare Reynolds: conception or design of the work, drafting the work or revising it critically for important intellectual content, final approval of the version to be published, agreement to be accountable for all aspects of the work.

## Funding

This study was funded by University College Dublin (UCD): Clare Reynolds, R21075.

## Keywords

cardiovascular dysfunction, developmental programming, gestational diabetes, maternal obesity, metformin, sex-specific

## Supporting information

Additional supporting information can be found online in the Supporting Information section at the end of the HTML view of the article. Supporting information files available:

**Peer Review History**

