## [Peer Review History · The Journal of Physiology]

Treated Mothers, Troubled Hearts: Cardiovascular Trade-Offs of Metformin Use in Pregnancy

Clare Reynolds

DOI: 10.1113/JP289360

Corresponding author(s): Clare Reynolds (clare.reynolds@ucd.ie)

Review Timeline:

Submission Date:	13-Jun-2025
Editorial Decision:	23-Jun-2025
Revision Received:	26-Jun-2025
Accepted:	30-Jun-2025

Senior Editor: Laura Bennet

Reviewing Editor: Christopher Lear

Transaction Report:

Dear Dr Reynolds,

Re: JP-P-2025-289360 "Treated Mothers, Troubled Hearts: Cardiovascular Trade-Offs of Metformin Use in Pregnancy" by Clare Reynolds

Thank you for submitting your manuscript to The Journal of Physiology. It has been assessed by a Reviewing Editor and an expert referee and we are pleased to tell you that it is acceptable for publication following satisfactory revision.

The review comments are copied at the end of this email.

Please address all the points raised and incorporate all requested revisions or explain in your Response to Referees why a change has not been made. We hope you will find the comments helpful and that you will be able to return your revised manuscript within 2 weeks. If you require longer than this, please contact journal staff: jp@physoc.org.

REVISION CHECKLIST:

We look forward to receiving your revised submission.

Yours sincerely,

Laura Bennet
Senior Editor
The Journal of Physiology

REQUIRED ITEMS

- The reference list must be in alphabetical order, rather than numbered, to comply with our Journal format.

EDITOR COMMENTS

Reviewing Editor:

Thank you for your well written perspective piece. Please see minor suggestions from the authors for your consideration.

REFEREE COMMENTS

Referee #1:

The authors have written an accurate and nuanced summary of the paper. There are no obvious misinterpretations or major scientific inaccuracies. I have a few minor comments for improvement - it is up to the author and editor's discretion whether these are incorporated.

1. "Insulin remains the standard treatment but there has been an increase in the use of the oral biguanide, metformin (Guo et al., 2019)." It may be useful to specify that the author is referring to global/worldwide trends here, as there are several countries including the UK where metformin is in fact the first-line pharmacological treatment after lifestyle changes are ineffective.
2. "However, subclinical effects were observed with increased cardiac output at 3 months and increased E/A ratio (increased blood entering the ventricle under filling) signalling a shift towards diastolic dysfunction". The diastolic dysfunction persisted with age and was progressive in nature, so for clarity a few words to explain this could be added at the end of this sentence to avoid suggesting that this occurred at 3 months only.
3. The Ob-Met males also showed vascular hyperreactivity to phenylephrine, which could be added to make the summary of the paper complete. However, I would understand if this was left out intentionally to simplify the conclusions of the work.

Many thanks to the author for writing this insightful and accurate perspective.

END OF COMMENTS

The authors have written an accurate and nuanced summary of the paper. There are no obvious misinterpretations or major scientific inaccuracies. I have a few minor comments for improvement - it is up to the author and editor's discretion whether these are incorporated.

1. "Insulin remains the standard treatment but there has been an increase in the use of the oral biguanide, metformin (Guo et al., 2019)." It may be useful to specify that the author is referring to global/worldwide trends here, as there are several countries including the UK where metformin is in fact the first-line pharmacological treatment after lifestyle changes are ineffective.

2. "However, subclinical effects were observed with increased cardiac output at 3 months and increased E/A ratio (increased blood entering the ventricle under filling) signalling a shift towards diastolic dysfunction". The diastolic dysfunction persisted with age and was progressive in nature, so for clarity a few words to explain this could be added at the end of this sentence to avoid suggesting that this occurred at 3 months only.

3. The Ob-Met males also showed vascular hyperreactivity to phenylephrine, which could be added to make the summary of the paper complete. However, I would understand if this was left out intentionally to simplify the conclusions of the work.

Many thanks to the author for writing this insightful and accurate perspective.

Response to Reviewers:

These are all excellent points and have been incorporated into the article. Thank you!

Dear Dr Reynolds,

Re: JP-P-2025-289360R1 "Treated Mothers, Troubled Hearts: Cardiovascular Trade-Offs of Metformin Use in Pregnancy" by Clare Reynolds

We are pleased to tell you that your paper has been accepted for publication in The Journal of Physiology.

Yours sincerely,

Laura Bennet
Senior Editor
The Journal of Physiology

If you would like to receive our 'Research Roundup', a monthly newsletter highlighting the cutting-edge research published in The Physiological Society's family of journals (The Journal of Physiology, Experimental Physiology, Physiological Reports, The Journal of Nutritional Physiology, and The Journal of Precision Medicine: Health and Disease), please click this link, fill in your name and email address and select 'Research Roundup':

<https://www.physoc.org/journals-and-media/membernews>

- You can help your research get the attention it deserves! Check out Wiley's free Promotion Guide for best-practice recommendations for promoting your work at: www.wileyauthors.com/eeo/guide. You can learn more about Wiley Editing Services which offers professional video, design, and writing services to create shareable video abstracts, infographics, conference posters, lay summaries, and research news stories for your research at: www.wileyauthors.com/eeo/promotion.

The Corresponding Author will receive an email from Wiley with details on how to register or log-in to Wiley Authors Services where you will be able to place an order

EDITOR COMMENTS

Thank you for your revised perspective

REFeree COMMENTS

Referee #1:

The author has addressed all comments to my satisfaction.